# Plant-Based Milk Alternatives in Child Nutrition

**DOI:** 10.3390/foods12071544

**Published:** 2023-04-06

**Authors:** Marco Brusati, Luciana Baroni, Gianluca Rizzo, Francesca Giampieri, Maurizio Battino

**Affiliations:** 1Pediatric and Educational Center “La Volpe e il Canguro”, 25062 Concesio, Italy; 2Scientific Society for Vegetarian Nutrition, 30171 Venice, Italy; 3Independent Researcher, Via Venezuela 66, 98121 Messina, Italy; 4Research Group on Food, Nutritional Biochemistry and Health, Universidad Europea del Atlántico, 39011 Santander, Spain; 5International Research Center for Food Nutrition and Safety, Jiangsu University, Zhenjiang 212013, China; 6Department of Clinical Specialistic and Odontostomatological Sciences, University Polytechnic of Marche, 60131 Ancona, Italy

**Keywords:** functional foods, plant-based formula, plant-based drinks, plant-based milk alternatives, child, infant, pediatric guidelines

## Abstract

Plant-based milk alternatives can be distinguished in two main categories, differing in production processes and regulation: plant-based formulas and plant-based drinks. They are now a widely accepted class of products on the international market. The various plant-based milk alternatives differ in nutritional characteristics due to their origin and manufacturing; more importantly, whereas formulas from plant and cow origin can be used interchangeably, plant-based drinks are nutritionally different from cow’s milk and can be consumed by children subsequently to the use of formula. Several scientific organizations have expressed differing opinions on the use of these products in the diets of children. In the face of unanimous conclusions regarding the use of these products during the first year of life, in subsequent ages there were conflicting opinions regarding the timing, quantities, and type of product to be used. From the viewpoint of the child’s overall diet and health, it could be suggested that these foods be considered not as simple substitutes for cow’s milk, but as part of a varied diet, within individual advice of use. We suggest accepting the presence of these products in a baby’s diet (omnivores included), planning their use correctly in the context of a balanced diet, according to the specific product and the needs of the individual.

## 1. Introduction

Although commonly called “milks”, all fluid products derived from a plant should not technically be called such. The term “milk” should be reserved for the fluid secreted from the mammary glands of mammals, thus speaking of mother’s milk, as in the case of human milk, cow’s milk, donkey’s milk, and so on [1,2,3,4,5,6]. In the case of products derived from a plant and that resemble, in their organoleptic characteristics, milks of animal origin, and even if the term “milks” is commonly used, it would be more correct to speak of “plant-based milk alternatives”.

The term “formula”, or “formulated or adapted milk”, should be reserved for products of animal or plant origin formulated for the specific needs of a child in the first years of life in the presence of physiological or pathological conditions.

In the context of the European Union, the regulatory framework for formulas is set by the Commission Delegated Regulation (UE) 2016/127 of 25 September 2015 [7], supplementing Regulation (EU) No 609/2013 of the European Parliament and of the Council [8].

All non-formulated plant-based alternatives (e.g., soy-, rice-, oat-based) have nutrient compositions, and therefore nutritional characteristics, that distinguish them from animal milks and formulas [9].

With the intent of simplifying the reading, in this paper we will refer to these products as “plant-based formulas” for the former and “plant-based drinks” for the latter.

Plant-based milk alternatives have been present in Asian tradition for centuries, but their diffusion in Western countries has less than a hundred years of history [10]. In recent decades, however, there has been a marked increase in demand, and consequently in supply, in Western countries as well, with an increase in market shares, to the disadvantage of cow’s milk [11,12,13,14]. In the absence of specific data in the pediatric age group, we can see that 77% of millennials in the United States make regular use of them [15], in Europe 15% of the population avoids cow’s-milk products [12], and in Italy, the market for plant-based drinks is growing (2016 data) with a +2% increase in soy-based drinks (small market variation, indication of its deep-rooted presence), whereas other “newer” plant-based drinks showed a more significant increase of 75.1% [11]. The reasons behind the choice of plant-based milk alternatives can be summarized in the following five main points:

1. With a prevalence of 2–3% of the population of less than 1 year of age, cow’s-milk protein allergy (CMPA) is the most common form of infant allergy, with a continuous reduction with advancing age, leading to a prevalence of less than 1% at 6 years [16]. As a result, the use of alternative formulas to cow’s milk is a mandatory choice in this group of subjects, at least initially, which can then be maintained over time [17].

2. Lactose intolerance refers to the clinical picture produced by the insufficient digestion of lactose by intestinal lactase. In addition to lactase deficiency in the first year of life, a more common entity is the non-persistence of the enzyme activity beyond the first year of life. This can lead to the appearance of symptoms of lactose intolerance, such as abdominal pain, bloating, flatulence, or diarrhea, which can manifest from 2 years of age [18,19] but more commonly from 5 to 6 years of age [19]. As plant-based alternatives (both formulas and drinks) are completely (and naturally) lactose-free [20], these products are proposed as alternatives to cow’s milk [18].

3. In cow’s milk, the presence of lipids, and in particular of saturated fatty acids and cholesterol, and on the contrary the presence of fiber, vitamins, and phytocompounds in plant-based drinks, can make families inclined towards the latter, with the aim of safeguarding their own health [17,21].

4. In the context of an increased intake of plant foods, which in general have a lower impact on the environment than animal foods [12], and that of an increased concern for animals used as a food source, plant-based milk alternatives can be part of a green and ethical choice for omnivores and lacto-ovo-vegetarian/vegans alike [5,14,21].

5. Finally, in the greater part of the world, milk of animal origin can be in short supply, expensive, or unsafe from a microbiological point of view, thus favoring products of plant origin [10].

In soy-based formulas, soy proteins are isolated from the other components of the soybean. In the final product, the minimum protein content is higher than that of cow’s-milk formula (2.25 g/100 Kcal vs. 1.8 g/100 Kcal) to compensate for the lower digestibility of plant proteins [22]. Moreover, there is a reduction of isoflavones, trypsin inhibitors, phytic acid, and fiber [9,23,24], and some components are added (iron, calcium, phosphorus, zinc, methionine, taurine, carnitine, arachidonic acid, docosahexaenoic acid) [22,24,25] to obtain a final product that satisfies the nutritional needs of an infant or child, according to specific regulatory criteria [26].

In rice-based formulas, the final product contains hydrolyzed rice proteins and a series of added compounds (lysine, threonine, tryptophan, carnitine, taurine, iron, zinc), with the same aim of rendering the nutritional composition suitable for the specific needs of a child [4,24,27]. In these formulas, the arsenic content of the rice from which they derive must not exceed 0.10 mg/kg [28]. Soy- and rice-based formulas are available with different formulations for children in the first year of life or older than 1 year of age [9].

Plant-based drinks derive from the respective plant through a series of industrial steps that include grinding (before or after soaking in water), homogenization (which makes uniform the dimensions of the particles present in the product), and separation of the solid phase from the liquid (filtration, centrifugation). The product is then subjected to treatments aimed at increasing its conservation (e.g., heat treatments such as pasteurization or UHT). Furthermore, during the various production steps, some substances such as vitamins, minerals, and stabilizers can be added in order to improve their nutritional, organoleptic, or stability characteristics [3,5,6,10,11,13,29,30]. Industrial processes can influence the nutritional characteristics of the final product, thus not always reflecting the characteristics of the plant of origin: e.g., regarding soy-derived drinks, a reduction of trypsin inhibitors and of the fiber oligosaccharide component is caused [6,30], and high-temperature treatments reduce the cholesterol-lowering effect of isoflavones [31]; however, not all researchers agree [32]. Regarding rice drinks, the regulatory limit of the arsenic content of the rice used in its manufacturing is 0.20 mg/kg [28].

A study based on 417 interviews showed that according to health professionals, half of consumers consider dairy products and plant-based alternatives to be nutritionally equivalent [33]. Among 215 parents who participated in an interview at the University of Miami and Jackson Memorial Hospital, 85% believed plant-based alternatives were at least equivalent to cow’s milk [34]. In another qualitative study conducted in Poland, Germany, and France, it was found that barriers and reinforcement to the consumption of plant-based alternatives depend on social factors, including health aspects, peer influences, and country-specific culinary traditions [35]. An online interview conducted in Denmark among young adults showed that consumers who consider milk and plant-based alternatives nutritionally equivalent are also those who consume them more often [36]. In a Canadian study based on perceptions and attitudes towards plant-based milk alternatives, plant-based drinks were associated with sustainability and health benefits among participants [37]. The above results highlight the importance of correct information for better acceptability. In the same direction, the recent Food & Drug Administration draft guidance on plant-based milk alternatives indicates that their goal “is to assist PBMA [plant-based milk alternative] producers in providing consumers with clear labeling to help them make more informed dietary choices when it comes to PBMA products” [38].

The purpose of this review is therefore to (a) summarize the nutritional characteristics of plant-based milk alternatives in comparison to cow’s milk; (b) review the guidelines on the use of plant-based milk alternatives in children, starting from birth; and (c) in light of these data, provide advice on the possibilities of using these products in specific situations.

## 2. Main Nutritional Characteristics of Plant-Based Milk Alternatives Compared to Cow’s Milk

Plant-based milk alternatives can be distinguished in two main categories, differing in composition, production processes, and regulation: plant-based formulas and plant-based drinks. For this reason, their nutritional characteristics differ between the two plant-based categories and, of course, are different from cow’s-milk characteristics.

### 2.1. Plant-Based Formulas

All formulas, both of plant and animal origin, are the final product of a manufacturing process that assembles single nutrients to obtain a food fitting the nutritional needs of the infant.

With regard to formula composition, the EFSA Scientific Opinion divides formulas into infant formula (IF) i.e., “food intended for use by infants during the first months of life and satisfying by itself the nutritional requirements of such infants until the introduction of appropriate complementary feeding,” and follow-on formula (FOF), i.e., “food intended for use by infants when appropriate complementary feeding is introduced and which constitutes the principal liquid element in a progressively diversified diet of such infant” [39]. Their main characteristics are summarized in Table 1.

For all formulas:The energy content is proposed to be between 60 and 70 Kcal/100 mL for IF and FOF [39]. In comparison, full-fat cow’s milk has 60 Kcal/100 mL [40].The allowed source of protein is cow’s milk, goat’s milk, isolated soy protein (ISP), and protein hydrolysates from “any suitable protein source and by different enzymatic or chemical means provided that the compositional criteria laid down by the Directive are met”. The minimum and maximum protein contents for IF and FOF in cow’s- and goat’s-derived formulas are 1.8 g/100 Kcal and 2.5 g/100 Kcal, respectively; for ISP 2.25 to 2.8 g/100 Kcal, respectively; and for protein hydrolyzates no minimum is set but the maximum is 2.8 g/100 Kcal. The Authors suggest a reference amino acidic pattern too [39]. As shown in Table 1, cow’s-milk protein content is 3.3 g/100 mL [40].Concerning the fat content, the EFSA panel proposes a fat content for IF and FOF that ranges between 40 and 55% of total energy, which is 4.4–6 g/100 Kcal of fat. The specific fat composition is detailed in particular for polyunsaturated fatty-acid requirements [39]. As a comparison, full-fat cow’s-milk fat content is around 3.2–3.7 g/100 mL [40]. In recent years, the European Union has made mandatory supplementation with docosahexaenoic acid for all formulas used in the first year of life [26].The carbohydrate content is calculated based on the residual energy after considering protein and fat composition. It can range between 9 g/100 Kcal and 14 g/100 Kcal for all kinds of formulas. Specific types of carbohydrates are detailed, including non-digestible carbohydrates [39]. As shown in Table 1, cow’s-milk carbohydrate content is 4.6 g/100 mL [40].The EFSA panel sets micronutrient composition for minerals and vitamins and discusses other ingredients such as choline, inositol, taurine, and probiotics, whose use can be planned in formulas. It is worth mentioning that higher amounts of iron are suggests for FOF compared to IF (minimum 0.6 mg/100 Kcal vs. minimum 0.3 mg/100 Kcal, respectively) when of animal origin and in protein hydrolyzates. ISP formulas should have a minimum iron content of 0.45 mg/100 Kcal for IF and 0.90 mg/100 Kcal for FOF [39]. In cow’s milk, iron is virtually absent [40].Finally, the EFSA panel does not consider it necessary to propose a specific composition of formulas used after 1 year of age, as “formula consumed during the first year of age can continue to be used by young children” [39]. The ESPGHAN position paper about young-child formulas (YCF) [41] agrees that follow-on formulas can be used after 1 year of age but calls for a “regulation of YCF to avoid inappropriate composition”.

### 2.2. Plant-Based Drinks

The nutritional composition of plant-based drinks can differ widely based on the plant source (grains, legumes, nuts and seeds, blends), manufacturing, and added ingredients. Here we focus mainly on soy-, rice-, oat-, and almond-based drinks, where not stated differently, as these appear to be the most diffuse worldwide [11,12,15,42]. Based on the above differences, it is crucial to choose a specific product to use. Their main nutritional characteristics are summarized in Table 2.

Plant-based drinks vary in energy content depending on the source of the product and the possible presence of added sugar. Usually, almond-based drinks are the lowest in calories but can vary widely (e.g., 26–46 Kcal/100 mL [11], 25–74 Kcal/100 mL [42]), whereas rice-based drinks are the richest ones (e.g., 54–61 Kcal/100 mL [11], 47–68 Kcal/100 mL [42]). Soy-based drinks have ~45 Kcal/100 mL [11,42]. Full-fat cow’s milk has ~60 Kcal/100 mL [40].Plant-based drinks vary in protein amount, too, as shown in Table 2. The most similar to cow’s-milk content (i.e., 3.3 g/100 mL [40]) are soy-based drinks (3.3 g/100 mL [11], 3.1 g/100 mL [42]), whereas rice-based drinks have the lowest protein content (~0.2 g/100 mL [11] and 0.3 g/100 mL [42]). Protein quality is different as well. Even though we have not found in the literature specific data about the quality of protein in plant-based drinks, it is known that protein of animal origin is of better quality when compared to plant protein due to the different aminoacidic profile and the lower digestibility [43]. However, it is possible to improve the quality of the diet through the careful planning of a single meal and the overall diet [44,45].Regarding the fat content, rice-based drinks usually have the lowest values of ~1 g/100 mL [11,42]. Soy-based drinks have ~2 g/100 mL [11,42]. Full-fat cow’s milk has ~3.2–3.7 g/100 mL [40], whereas partially skimmed and fat-free cow’s milk have lower fat content. There are differences in the fatty-acid composition: a prevalence of saturated fatty acids is typical of cow’s milk (around 60%) and coconut-based drinks (around 90%) [10,36], whereas a prevalence of polyunsaturated fatty acids is reported for soy-based drinks [36]. Moreover, during the production, different oils (such as sunflower, rice) can be added to improve the organoleptic characteristics, making the lipid profile of the final product different from what could be inferred considering only the primary source.Rice-derived drinks usually are the richest in carbohydrates, with values slightly over 10 g/100 mL (12 g/100 mL [11], 11.5 g/100 mL [42], followed by oat-based drinks (7 to 8 g/100 mL) [11,42]. Soy-based drinks usually have a carbohydrate content ranging from 3 g/100 mL [11] to 4 g/100 mL [42]. Cow’s milk sits around 4.6 g/100 mL [40]. The main sugar in cow’s milk is lactose, whereas plant-based drinks are naturally devoid of lactose. However, sugars can be added to plant-based drinks, so the nutritional composition can vary widely according to sources and manufacturing.Whereas calcium is naturally present in cow’s milk (on average, 120 mg/100 mL), it can be added to plant-based drinks. Moreover, calcium bioavailability may differ depending on the type of fortification used (tricalcium phosphate, calcium carbonate, red alga *Lithotamnium calcareum*), even though the 2016 Academy of Nutrition and Dietetics position paper on vegetarian diets reports that calcium absorption “from most fortified plant milks is similar to that from cow’s milk, at approximately 30%” [46].Vitamin D, vitamin B2, and vitamin B12 can also be added to plant-based drinks, depending on the choice of the manufacturer. In the case of vitamin D, vitamin D3 can be of animal origin (from sheep lanolin), but in the last few years botanical sources of vitamin D3, *Cladonia raingiferina* (reindeer lichen), have become available [47].

## 3. The Role of Plant-Based Milk Alternatives in the Diets of Children 0–12 Months

### 3.1. Allergy to Cow’s-Milk Protein

Cow’s-milk protein allergy (CMPA) is a pathological situation in which the subject develops an allergy to components of cow’s milk, as well as to other mammals’ milks, given the frequent cross-reactivity between the milk proteins of different mammals. It is a problem that affects 2–3% of children in the first year of life and whose prevalence then decreases to drop below 1% at 6 years of age.

The immune response can be Ig-E mediated in 60% of cases, whereas the remaining 40% can be Ig-E negative or consist of mixed forms [48]. Though a consistent overlap exists, IgE-mediated forms usually consist of typical immediate reactions (from skin, respiratory, gastrointestinal, or cardiovascular reactions to anaphylaxis), whereas non-IgE-mediated forms can manifest as food-protein-induced enterocolitis syndrome (FPIES) or food-protein-induced proctitis/proctocolitis (FPIAP) [48]. Lastly, mixed forms can display an IgE and/or eosinophilic component and manifest, for example, as eosinophilic gastrointestinal disorders [48,49].

The possible severity of the situation and the specific nutritional needs related to the age and type of nutrition of children in the first 12 months of life require careful dietary management. The specific suggestions are summarized in Table 3.

As it is not possible to use a standard formulated milk based on cow’s-milk proteins, the guidelines of the World Allergy Organization (WAO), the American Academy of Pediatrics (AAP), and the European Society of Pediatric Gastroenterology Hepatology and Nutrition (ESPGHAN) recommend using formulas with hydrolyzed cow’s-milk proteins (extensively hydrolyzed formula—EHF), for which the efficacy in terms of normal growth and non-allergenicity has been ascertained in 90% of children [16,25,50]. In the case of specific conditions (e.g., anaphylaxis, severe enteropathy, multiple allergies, or symptoms not completely resolved by EHF), it is recommended to use formulas based on isolated amino acids (amino-acid formula—AAF), essentially devoid of sensitizing potential [16,50], as shown in Table 3. Two possible alternatives are represented by rice-based and soy-based formulas. The former can be used as a first-line treatment [9] and in case of problems with EHF [16,22]. The latter may present a risk of cross-reactivity with cow’s-milk proteins in 10–14% of children: Their use is taken into consideration in the second semester of life (not in the first), once the tolerance to soy proteins has been established, in the presence of problems with hydrolyzed or amino-acid formulas [9,16,22], and in the absence of gastrointestinal symptoms [24]. The North America Society of Pediatric Gastroenterology Hepatology and Nutrition (NASPGHAN) position paper of 2020 also agrees with the use of infant formulas in the case of CMPA, including soy-based ones [51]. In the case of food-protein-induced enterocolitis syndrome (FPIES), the AAP does not recommend the use of soy-based formulas [25].

Given the necessity to respect the needs of this age group, plant-based drinks, without a specific formulation for age and pathological condition, should not be used in these subjects [9,16,51].

### 3.2. Galactosemia and Lactose Intolerance

Galactosemia refers to an inborn error of carbohydrate metabolism that makes the subject unable to metabolize that sugar. This aldohexose is metabolized via different enzymatic steps that can be altered, giving rise to the disease [52]. The incidence of the disease varies greatly, with estimates ranging from 1 in 480 births to 1 in 60,000 births [53].

In the congenital form of lactase deficiency, the newborn is unable to digest lactose to glucose and galactose. This is a very rare disease, as only a few cases have been described, mostly in Finland, probably because of a founder effect [18].

Both of these disorders, as well as severe damage to small-intestinal mucosa, are indications for the use of lactose-free infant formulas, including formulas based on isolated soy proteins [22,25,54,55,56].

If, on the other hand, lactose intolerance is less pronounced, as could be supposed for infantile colic due to a transient low lactase activity [19], a trial of a low-lactose formula may be carried out, but reduced-lactose formula or lactose-free formula are not suggested as a routine approach in these situations [19,25].

### 3.3. Preterm Infants

The two consensuses of ESPGHAN [22] and AAP [25] agree that soy-protein formulas should not be used in preterm infants [22] due to the presence of scarce or detrimental information in this group of subjects. Of the same opinion are the Australian Ministry of Health [55] and the review by Vandenplas et al. [57].

### 3.4. Choosing Plant-Based Milk Alternatives for Family Preferences

The feasibility of using formulas based on isolated soy proteins (or rice hydrolyzates [58,59]) in the case of parent preference (e.g., cultural, religious, or ethical reasons, or vegan families) is recognized and considered acceptable and appropriate by the ESPGHAN [22]; the AAP [25]; the French-speaking Group of Pediatric Hepatology, Gastroenterology and Nutrition (GFHGNP) of the French Society of Pediatrics [59]; the Spanish Society of Pediatrics [60]; the European Pediatric Association [61]; the Norwegian Nutrition Council [62]; the Australian Ministry of Health [55]; and the Canadian Ministry of Health [54]. Other publications agree with this approach [44,58,63]. Of the same opinion are the AAP Pediatric Nutrition Manual [64,65] and the manual by Mangels et al. [66].

The position paper of the German Nutrition Society, on the other hand, has a different opinion, which advocates for the use of soy-based formulas only in “exceptional and justified cases (e.g., galactosemia) and on medical recommendation” [56].

It is important to note that the vitamin D that fortifies infant formulas is generally vitamin D3 of animal origin (from sheep lanolin). Therefore, technically even soy- or rice-based formulas are not vegan [66]. However, looking at botanical sources of vitamin D3, *Cladonia raingiferina* (reindeer lichen) is a known source of vitamin D3 and vitamin D2 at relative high levels (67–204 and 22–55 µg/100 g of dry matter, respectively), along with ergosterol and 7-dehydrocholesterol [47]. It is already exploited as a source of vitamin D3 in commercially available supplements, so we speculate that in the future this problem may be solved.

The NASPGHAN position paper on the use of plant-based milk alternatives in the first year of life stresses that the source of milk at that age must be “human milk or an iron-fortified infant formula”, since a large part of the needs are met by milk at this age [51]. The Health Ministries of New Zealand [67], Australia [55], and Canada [54] also advise against the use of plant-based drinks during the first year of life. The same opinion is also expressed in the Healthy Eating Research publication [68]; that of the GFHGNP of the French Society of Pediatrics [59]; those of Mangels et al. [69], Baroni et al. [58], and Sethi et al. [30]; and the Handbook of Pediatric Nutrition of the AAP [70].

On the contrary, as regards the use of plant-based drinks as ingredients in the preparation of complementary foods (similarly to vegetable broth or water) for use during complementary feeding and not as substitutes for breast milk or formulas, the approval of this possible use can be found in recent publications [44,58,66,71].

## 4. The Role of Plant-Based Milk Alternatives in the Diet of Children over 12 Months of Age

### 4.1. Allergy to Cow’s-Milk Proteins

In the case that CMPA persists or occurs beyond 1 year of age, the guidelines of ESPGHAN [16], NASPGHAN [51], the Mexican Association of Gastroenterology (AMG) [6], and other recent publications [9,24] continue to recommend a formula (EHF, AAF, soy- or rice-based formulas) as the first choice to ensure the nutritional needs in a cow’s-milk protein-free diet. On the other hand, the conclusions of Healthy Eating Research are slightly different: “Between 1 and 5 years of age, plant-based drinks can be particularly useful for children with allergies or intolerances to cow’s milk” [68], thus opening to the use of plant-based drinks in these situations after 1 year of age.

### 4.2. Lactose Intolerance

The non-persistence of lactase activity beyond childhood is a very common situation in which a part of the world population cannot digest lactose, with a prevalence that varies among ethnic groups from 15 to 100% of the adult population [18].

In this situation, the advice is to reduce the intake of lactose, and only more rarely is a completely lactose-free diet necessary [19]. As regards milks, Berni Canani et al. recommend lactose-free milks or soy-based drinks [18].

### 4.3. Choosing Plant-Based Milk Alternatives for Family Preferences

The position of the Mexican Association of Gastroenterology (AMG) is summarized in a recent publication, with specific reference to soy-based drinks [6]. The authors conclude by stating that “there is no evidence on the health benefit of plant-based drinks in childhood nutrition” and that plant-based drinks “should not be utilized as a replacement for breastmilk or as a replacement for breastmilk substitutes in the feeding of children during the first 2 years of life. Their later use as part of the liquid portion of diet must be individualized”. They then add that in the vulnerable segments of the population (children, adolescents, the elderly) plant-based drinks should be fortified and included in the context of a balanced diet [6]. The position of NASPGHAN is to recommend it up to 2 years of age, apart from breast milk, cow’s milk, or formulated milk. The position of ESPGHAN [41] regarding the use of infant formulas in the 12–36-month age range (young-child formula—YCF) focuses on formulated milks of animal or plant origin specifically designed for this age range and therefore does not deal with plant-based drinks. The conclusions are nevertheless interesting for the focus of our study: “There is no necessity for the routine use of YCF in children from 1 to 3 years of life, but they can be used as part of a strategy to increase the intake of iron, vitamin D, and n-3 PUFA and decrease the intake of protein compared to unfortified cow’s milk. Follow-on formulae can be used for the same purpose. Other strategies for optimizing nutritional intake include promotion of a healthy, varied diet, use of fortified foods, and use of supplements”. Furthermore, “the protein content should aim toward the lower end of the permitted range [for 6–12 months formulas] if animal protein is used”, i.e., towards 1.6 g of protein/100 Kcal (about 1.1 g/100 mL) [41]. These statements are in line with the findings of the European Food Safety Authority [39] and taken up by the European Commission in 2016 [72] in the sense of not being able to identify a unique role for YCF, as other nutritional strategies can achieve the same results. The American Academy of Nutrition and Dietetics (AND) in its 2016 position paper on vegetarian diets stated that fortified soy-based drinks or cow’s milk can be used from 1 year of age in children growing normally and consuming a variety of foods [46]. The British Dietetic Association (BDA) Food Fact Sheet briefly states that “from the age of one year, fortified plant-based drinks can be used in preparing foods and given as the main milk drink” [73]. In the recent Guidelines for a Healthy Diet of the Italian Food and Nutrition Research Center [74] there are no references to plant-based drinks. Other individual groups have also taken positions on the use of plant-based drinks in children. Wright and Smith believe that “after weaning, typically around 12 months of age, milk of any kind is not required, that children will be fine with water and a good healthy balanced diet”, and that plant-based drinks “can easily be included in day-to-day use after weaning, and, used in this fashion, plant milks can be less harmful than dairy milks, at least for certain groups” [75]. For Craig et al., from 1 year of age, if the child grows normally and consumes a variety of foods, fortified plant-based drinks derived from soy or peas can be introduced in addition to cow’s milk [76].

#### Adequacy of Plant-Based Drinks Compared to Cow’s Milk

A plant-based drink, which is not substantially equivalent to cow’s milk, is not considered an adequate alternative to cow’s milk under 2 years of age [51]. For the Spanish Society of Pediatrics, calcium-fortified plant-based drinks “should never be used as the main liquid food of the child, at least until age 2–3 years” [60]. The GFHGNP of the French Society of Pediatrics, speaking about iron requirements for a child who follows a vegan diet, recommends soy- or rice-based formulas for “as long as possible, ideally up to 6 years of age” [59]. The advice of Healthy Eating Research [68], expressed in “Healthy Beverages Consumption in Early Childhood”, signed by the Academy of Nutrition and Dietetics, the American Academy of Pediatric Dentistry, the American Academy of Pediatrics, and the American Heart Association, is that between 1 and 5 years of age plant-based drinks be used in case of “specific dietary preferences”—and mentions in particular the vegan and lacto-ovo-vegetarian diets—in addition to medical conditions that may require it. However, they recommend that their consumption is not exclusive (except in the case of soy-based drinks) and that in any case the choice should be evaluated with a practitioner to monitor the adequacy of the whole diet [68].

The recent guidelines of the New Zealand Ministry of Health report the possibility of using soy-based drinks from 1 year of age, provided they are fortified with calcium (and vitamin B12 if the child follows a vegan diet and does not receive supplements); rice- and other grain-based drinks are not recommended as the sole substitute for cow’s milk up to 5 years of age, and fortified types are generally recommended [67,77]. For children following a vegan diet, the Australian Ministry of Health guidelines recommend continuing with a soy-based formula up to 2 years of age. Indeed, the same guidelines state that fortified soy-, rice-, or oat-based drinks “can be used after 12 months under health professional supervision” [55]. The Canadian joint statement—supported by Health Canada, the Public Health Agency of Canada, the Canadian Pediatric Society, and the Dietitians of Canada and Breastfeeding Committee for Canada—suggests the use of soy-based formulas up to 2 years of age if the child does not take cow’s milk. In addition, it states that “soy-, rice-, almond- or other plant-based milks [...], whether or not they are fortified, are not appropriate as the main source of milk for a child younger than two years” [54].

Mangels et al., in a handbook on vegetarian nutrition [78], leave freedom of use for whole-cow’s-milk or soy- or pea-based drinks that are fortified and unflavored from 1 year of age, provided that the child—if vegan—grows regularly and has reliable sources of iron and zinc in the diet. In addition, given the lower lipid content of soy- and pea-based drinks compared to whole cow’s milk, they recommend the presence of other sources of lipids in the diet up to 2 years of age, when fats should not be limited in the diet of the child. Due to the different nutritional composition, the guide does not recommend other plant-based drinks as the main drink for young children [78].

For Vandenplas et al., these plant-based drinks “should ideally not be used as a main drink in children <2 years of age, and if they are considered after 1 year of age, nutritional assessment should occur before, to ensure that the child is achieving their nutritional requirements through their current diet” [9]. For Verduci et al., all plant-based drinks “should not be used as a substitute for cow’s milk in children <24 months old” [24]. For Sethi et al., plant-based drinks are “inappropriate alternatives for breast milk, infant formula or cow’s milk in the first 2 years of life,” and beyond this age, when consumed for medical reasons, they recommend fortified products that contain at least 6 g of protein in 250 mL of product [30]. On the contrary, for Mangels and Driggers, fortified soy-based drinks are an “appropriate substitute for cow’s milk” from one year of age; the other alternatives should be used occasionally given the lower protein and energy content [69].

## 5. Advice Common to All Choices

### 5.1. Age

A substantial uniformity of views is derived from the reviewed literature regarding the use of plant-based formulas and plant-based drinks in the first year of life. The former are considered adequate in specific situations [7,14,20,22,23,32,33,34,35,38,39,40,44,46], whereas the latter’s use is suggested for the preparation of foods, once complementary feeding has begun, but not as a substitute for breast or formulated milk [9,16,51,54,55,58,59,67,68,69,70].

On the contrary, the conclusions regarding later ages are less unanimous. As regards the age from which plant-based drinks can be introduced as an alternative to cow’s milk, for some authors those derived from soy can be introduced at 1 year of age [46,67,68,76,78]. Other authors speak more generally in favor of the use of all plant-based drinks from 1 year of age [55,73,75]. Furthermore, others speak of occasional use from 1 year of age of plant-based drinks other than soy [67,68,69]. Finally, some authors reserve the use of plant-based drinks, as a group, for no earlier than 2 years of age [6,24,51,54,59,60]. The interpretation of the “suitable” age provided by the AAP Pediatric Nutrition Manual is that plant-based drinks should not be used as a substitute for breast milk or formula, i.e., when these “provide a significant portion of daily energy intake” [70], which leaves room for the interpretation of the term “significant” but lays the foundations for a personalized solution to the problem.

### 5.2. Quantity

In the first months of life, the amounts of human milk or formula (no matter the origin) can be calculated on a per-kg basis, i.e., 140 to 200 mL/kg body weight per day [64]. Moreover, “when formula feeding is used, bottles should be offered ad libitum, the goal being to allow the infant to regulate the intake to meet his or her energy needs” [64]. Growth of the child, bowel habits, urine output, general behavior, and possible symptoms have to be monitored over time.

When complementary foods are introduced into the diet, breast milk or formula continue to be a substantial part of the diet of the infant but a reduction in the milk intake (breast milk and formula) is supposed to occur; during this period, responsive feeding practices are suggested, but in the context of a comprehensive evaluation of the infant’s growth [70].

When plant-based drinks can be introduced, it is a common practice that families use those drinks and cow’s milk interchangeably—even though they differ in processing and final characteristics. Indeed, although the suggested amounts of plant-based drinks are rather uniform, the proposed amounts of cow’s milk are different, as shown in Figure 1.

With regard to the recommended amount of cow’s milk and formulas after the first year of age, the different guidelines provide different references. Suthutvoravut et al. report values of 200–400 mL of formula per day for the 12–36-month range [79]. The Italian CREA guidelines of 2018 [74] recommend 150 mL of cow’s milk in the 1–2-year range and from 2–3 years of age onwards 200 mL per day; the F.A.Q. of the Italian Ministry of Health in “Proper nutrition and nutritional education in early childhood” report 200–400 mL of cow’s milk per day “after the year of life” [80]; Verduci et al. recommend 150 mL of whole cow’s milk or 300 mL of YCF in the 12–24-month range [81]; the EFSA recommends 300–500 mL of cow’s milk after the first year of life in “young children” [82]; the New Zealand guidelines report values of 350 mL of cow’s milk from 12 months of age [67] and up to 500 mL of milk from 2 years of age [77]; the ESPGHAN position paper on the iron status in children recommends quantities of cow’s milk of less than 500 mL per day from 1 to 3 years of life [83], values in line with the indications of the Early Nutrition Project for the second year of life [84]; and the recommendations of Healthy Eating Research are 500–700 mL of cow’s milk at 12–24 months, a maximum 500 mL at 2–3 years, and a maximum 600 mL at 4–5 years [68]. The amount of cow’s milk suggested by the various references are summarized in the second and third columns of Figure 1.

Concerning plant-based drinks, their consumption is seen in the context of the overall diet of the child as a function of the total intake of calcium and protein, and therefore have less stringent indications on the recommended-quantity values for the individual product, as shown in the fourth and fifth columns of Figure 1. Thus, if Messina and Mangels recommend 3 cups of soy-based drink in vegan children (from toddlers to school-aged children) [44], lower values (3/4 of cup of soy-based drink or as much cow’s milk) can be found in the typical 2-year-old’s diet, which also includes the consumption of cheese and yogurt of animal or plant origin [69]. In line with this approach, in “The Dietitian’s Guide to Vegetarian Diets”, Mangels et al. recommend 2 cups per day of cow’s milk or calcium-fortified plant-based drinks, with which other sources of calcium are combined [78]. The AND in its “Resources for consumers” recommends yogurt and cheese as well as 1.5 cups of soy-based drink for a 2–3-year-old lacto-ovo-vegetarian child [85]. Other authors [86,87] consider a serving of milk to be equal to 200 mL, but in this case the value, rather than an indication of the recommended daily intake, has the characteristics of an operational reference, as both groups develop a diet-planning system that looks at the total food intake and not just at the single food.
Figure 1Graphical representation of the suggested amount of cow’s milk and plant-based drinks for children aged 12 to 24 months and from 24 months of age onward, according to the selected literature—see text for details (1 cup = 236 mL) [44,67,68,69,74,77,78,80,81,82,83,84,85,86,87].
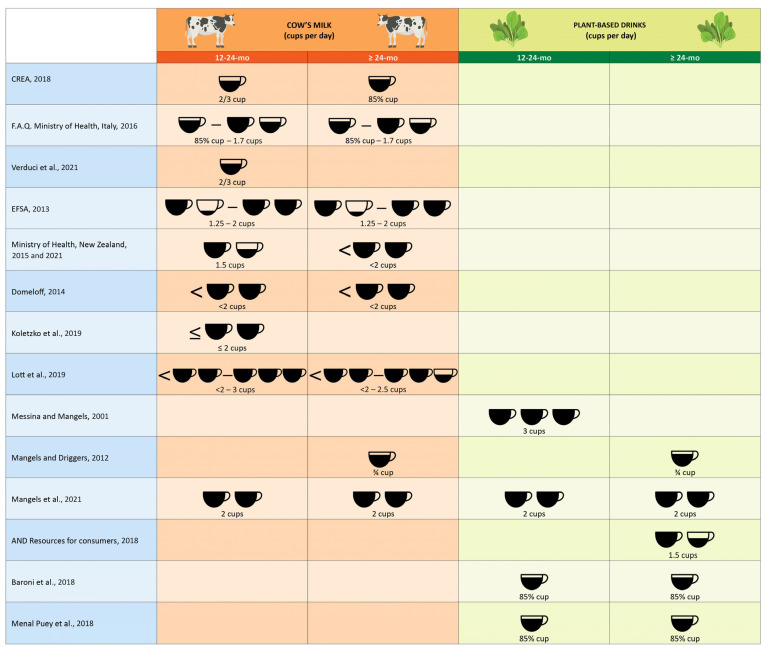


## 6. Advice in Specific Situations

This section provides advice for specific situations to allow the better option of plant-based milk alternatives to be chosen. This must be considered with the overall diet and adapted case by case. The suggestion of avoiding plant-based drinks with added sugar or sweeteners and preferring calcium-fortified products is still valid. In order to better focus on and summarize the concepts, we summarized the information in Table 4.

### 6.1. Overweight, Obesity, Dyslipidemia

In these conditions, in children over 1 year of age, it may be useful to use a soy-based or almond-based drink [21,88], which is less energetic. In both cases, the fat content is lower than that of whole cow’s milk (although higher than that of skim milk) [2,3,5,11,12,13,17,42], but, above all, their composition is different, with a lower total and percentage amount of saturated fatty acids (in favor of controlling the levels of cholesterol) and a greater proportion of energy from mono- and polyunsaturated fatty acids [14,15,21]. The overall energy intake is also lower, especially with regard to almond-based drinks and compared to the content of whole cow’s milk (as well as to that of partially skimmed milk) [2,3,5,11,12,13,15,17,42]. Soy-based drinks also have a composition (especially protein and isoflavones) that seems to have a positive impact on blood lipids [89,90]. Even an oat-based drink rich in β-glucans could be useful for controlling cholesterol levels [24,30,91]. Moreover, recent data suggest that spelt-based drinks may have an important satiating effect [92]. Furthermore, as in the case of diabetes, limiting the intake of any plant-based drink, which can cause large fluctuations in blood sugar and insulin (rice in particular [20]), could be beneficial. Table 4 summarizes these indications in the relative line. If the family’s preferences fall on cow’s milk, advice might be given to limit its intake to no more than 150–200 mL per day, thus limiting the intake of energy and protein (also considering the association between excessive intake of protein in the first two years of life and subsequent obesity [93]). For the lipid component, a partially skimmed cow’s milk could be recommended as early as the second year of life [70,94]; an alternative could be animal- or plant-based formulas for 12–36-month-old children.

### 6.2. Diabetes Mellitus

Beyond 1 year of age, the second line of Table 4 shows that in the presence of risk factors for both type 1 and type 2 diabetes mellitus, or if the disease is overt, a limitation of the intake of rice-based drinks, rich in carbohydrates and simple sugars and with a high glycemic index and glycemic load [20], could benefit glycemic control, especially avoiding significant fluctuations in glycemia and consequently in insulin. Soy-based drinks, thanks to their low supply of carbohydrates and sugars (but it is necessary to check that there are no added sugars in the individual product) and with a lower glycemic index, could be a valid alternative in this sense [14,15,89]. Oat-based drinks have an intermediate glycemic index [20], and, due to the action of β-glucans, they could have positive effects on glycemia and insulin response [10,30,91].

### 6.3. Poor Weight and Stature Growth

In the case of a child who has a poor weight and/or stature growth after the first year, as seen in the third line of Table 4, the choice could be based on products with a higher energy and nutrient density, such as cow’s milk or soy- or rice-based drinks [2,3,5,11,12,13,15,17,20]. For a child with limited growth in height, the choice could be to opt for cow’s milk, for which the association between its consumption and higher height is known [95,96]. However, these suggestions are to be applied in the context of dietary indications concerning the overall diet of the child.

### 6.4. Allergy to Cow’s-Milk Proteins

As seen previously (Table 1), when this disease occurs during the first year of life, the possible therapeutic alternatives to a normal formula derived from cow’s milk are a formulated milk of cow origin with extensive hydrolysis of proteins or an amino-acid formula, or a plant-based formula from isolated soy proteins or rice hydrolyzates—a decision that will be made in accordance with the team that manages the child’s disease. If the allergy persists beyond the first year of age or occurs in later ages, in addition to the possibility of continuing up to 24 months (and beyond) with adapted formulas [6,9,16,24,51], the possibility of using plant-based drinks may also be considered [68], as shown in the fourth line of Table 4.

### 6.5. Lactose Intolerance

If lactose intolerance is present in the first year of life, formulas for infants with reduced lactose content can be used in the case of non-severe manifestations [25], but if the manifestations are severe, lactose-free formulas, which include formulas based on soy-protein isolate, can be used [22,25,54,55,56]. If the problem develops after 1 year of age, a plant-based drink—naturally lactose-free—can be an alternative to lactose-free cow’s milk [18] (Table 4, fifth line).

### 6.6. Lacto-Ovo-Vegetarian and Vegan Diet or Family Preference for Plant-Based Milk Alternatives

In the first year of life, in the presence of a request for a formulated milk of non-animal origin, almost all of the authors foresee the possibility of using a formula based on soy proteins or a rice hydrolyzate [22,25,44,54,55,58,59,60,61,62,63,64,65,66]. For a preterm infant, there is currently insufficient available data about plant-based formulas to allow their use, and therefore it seems prudent to direct the family towards the choice of breast milk from a milk bank, where possible, or a formulated cow’s milk [25,55,57,97]. In the context of complementary feeding, plant-based milk alternatives can be used in food preparation [44,58,66], with the foresight of limiting (or avoiding) the use of rice-based drinks due to the possible arsenic content [15,98]. Starting from one year of age, in the case of vegetarian or omnivore subjects, there is the possibility (a) to continue with the formulas used during the first year [41], (b) to choose the formulas adapted from cow’s milk +12 months (young-child formula) [41,81], or (c) to use cow’s milk as such, generally whole up to 2 years of age and then partially or totally skimmed [46,68,70,81]. Alternatively, from 1 year of age [9,46,55,67,68,69,73,75,76,77,78] or from 2 years of age [6,24,30,54,60], plant-based drinks can be used by both lacto-ovo-vegetarians and omnivores and will be the product consumed by children following a vegan diet (last line of Table 4). A plant-based alternative after the first year of life can also be the use of plant-based formulas adapted for the 0–12- or 12–36-month range [54,55,59]: Since these are nutritionally similar to cow’s milk, their use from 1 year of age can be more intuitive than that of a plant-based drink.

Even if cow- or plant-based formulas can be used on the basis of an infant’s weight in the same way, this procedure is usually not used after 6 months of age [64,70]. In any case, regardless of the dietary pattern followed, the milk—in the broad sense—chosen must be considered in the context of the child’s overall diet and according to the product: Adequate counseling regarding the choice and use of these products will therefore be essential.

Therefore, considering the breakfast of a healthy child 1 year of age or more (a meal in which typically, but not necessarily, milk is present), if the family chooses to use a plant-based drink instead of cow’s milk, a series of general advice can be proposed as examples. Such advice should be personalized according to the rest of the diet, the single product chosen, the growth parameters of the child, and the possible use of supplements.

Perhaps with the exception of soy-based products, it may be advisable, especially in the second year of life, that a single source be not the only plant-based drink used to replace cow’s milk. It is suggested to offer plant-based drinks from different sources, alternating them during the week(s). This, among other factors, could add variety to the diet and encourage the acceptance of different foods in younger children.Due to the saturated-fat content of coconut-based drinks, these products are generally not recommended in children’s diets, except for occasional use.If a family chooses plant-based drinks from less common sources (flax, hemp, drinks with multiple components), it is essential to consider the nutritional information of the single product, always in the context of the individual’s overall diet.The basic breakfast may respect the general setting [74,97], thus providing a carbohydrate source, a protein source, a fat source, and a source of fiber, minerals, and phytocompounds—with milk as the source of protein and fat.

We tried to summarize the following general suggestions in Table 5 for better comprehension.

#### 6.6.1. In the Case of Using Plant-Based Drinks Derived from Soy

It is suggested to choose drinks enriched with calcium (and possibly other minerals and vitamins) and free of added sugars.According to the CREA indications [74], for the 12–24-month age group, 150 mL of whole cow’s milk provides 90 Kcal, a value in line with the 80 Kcal per 200 mL of soy-based drink. One hundred fifty mL of whole cow’s milk provides 4.8 g of fats, a value slightly higher than that provided by 200 mL of soy-based drink (about 4 g) [40]. In this regard, beneath considering a possible additional source of fats in the meal, it should also be pointed out that the AAP allows the use of “low-fat” cow’s milk starting from the second year of life not only in a child with excess weight but also in a regularly growing child [70]. As for the protein intake, 200 mL of soy-based drink provides 5.6–7 g of proteins versus 5 g of proteins from 150 mL of cow’s milk [40]: This difference should not be a problem given the lower digestibility of plant protein.After the first 2 years of life, the official recommendation is that of switching from whole cow’s milk to partially or totally skimmed milk, with 200 mL per day [74]. About 100 Kcal are derived from 200 mL of 2% fat cow’s milk and 80 Kcal from 200 mL of soy-milk. We can consider the contributions of fat and protein to be roughly similar to that of cow’s milk: 200 mL of soy milk has about 4 g of fats, the same value as 200 mL of cow’s milk with 2% fat; regarding proteins, the intake from 200 mL of soy-based drink is 5.6–7 g, similar to 6.6–6.8 g per 200 mL of cow’s milk [40].

#### 6.6.2. In the Case of Using Plant-Based Drinks Derived from Grains

Limit the consumption of rice-based drinks in the first years of life (more or less for the first five years) due to the possible arsenic content.It is suggested to choose grain-based drinks enriched with calcium (and possibly other minerals and vitamins) and free of added sugars.Due to the carbohydrate content, sugars in particular, it may be advisable not to use rice-based drinks, and limit oat-based drinks, in a breakfast already rich in simple sugars (e.g., in a breakfast that includes bread and jam, biscuits, or cake prepared with added sugar). On the other hand, if we consider an older child or an adolescent who may exercise in the morning, grain-based drinks could be a choice to be considered.As for the 12–24-month range, 150 mL of whole cow’s milk provide 90 Kcal, 5 g of proteins, and 4.8 g of fats. Considering 200 mL of a grain-based drink, the contributions are similar in terms of calories (about 100 Kcal from 200 mL of rice- or oat-based drink) but different for proteins and lipids: For rice-based drink the same amount provides 0.6 g of proteins and 2 g of fats, and for oat-based drink 1.6 g of proteins and 5.4 of fats [40]. Therefore, it would be useful to provide another source of protein at breakfast if one of the above grain-based drinks is consumed, and another source of lipids in the case of rice-based drink.As for children 2 years of age and older, 200 mL of 2% fat cow’s milk provide 100 Kcal, 6.6–6.8 g of proteins, and 4 g of fats. As described above, the same amount of rice-based drink provides 100 Kcal, 0.6 g of proteins, and 2 g of fats; for 200 mL of oat-based drink the values are 100 Kcal, 1.6 g of proteins, and 5.4 g of fats [40]. Although the intakes of calories and lipids are similar, it would be useful to provide another source of protein in the case of grain-derived drinks.

#### 6.6.3. In the Case of Using Plant-Based Drinks Derived from Almonds

It is suggested to choose almond-based drinks enriched with calcium (and possibly other minerals and vitamins) and free of added sugars.Compared to the values of 150 mL of whole cow’s milk (90 Kcal, 5 g of proteins, and 4.8 g of fats), 200 mL of almond-based drink provide 30–40 Kcal, 1.2–1.4 g of proteins, and 2.4–3.2 g of fats [40]. It would therefore be advisable to include another source of protein and fat in the meal.Compared to 200 mL of semi-skimmed milk, the intake of 200 mL of almond-based drink remains lower in calories (100 Kcal vs. 30–40 Kcal, respectively) and proteins (6.6–6.8 g vs. 1.2–1.4 g) but similar in terms of fats (4 g vs. 2.4–3.2 g) [40]. Therefore, it would be useful to include another source of protein in the meal.

## 7. Limitations

Even if plant-based drinks are recognized in the literature as milk substitutes, they have different biological values and nutritional characteristics than cow’s milk, so these two types of foodstuffs cannot overlap. The content of some critical vitamins and minerals, such as calciferol, riboflavin, iron, zinc, and calcium, may vary widely, especially in these drinks.

However, it is crucial to consider fortification as an effective approach to adapting plant-based products to cow’s milk. This approach is already used for infant formulas (which are commonly accepted regardless of plant or milk origin). Moreover, even if the protein quality of single-plant food can be low, the protein quality of the meal and the overall diet can be increased through the variety of foods eaten in the same meal and throughout the day.

A limitation of these recommendations is that of having used the USDA database (Foundation Foods and Standard Reference Legacy Foods) as a reference for the average intakes of the different types of plant-based drinks. In fact, the values reported may not be representative of the individual product’s reality, given the wide (and always increasing) offer on the market. In addition, we only considered plant-based drinks with no added sugar (as recommended) despite there are also being products that contain added sugars and therefore have an impact on the intake of energy and the overall quality of the diet. However, we repeated the same calculations starting from unpublished personal data concerning products available on the Italian market, including both plant-based drinks with and without added sugars, obtaining substantially comparable results. The suggestion is therefore that, starting from these general indications, the professional evaluate with the family the products they intend to use to provide tailored information for the specific situation.

## 8. Conclusions

Although the literature concerning the first year of life is substantially in agreement in recommending the use of formulas while limiting the use of plant-based drinks to the preparation of complementary foods, the indications after 12 months of age are less homogeneous both in terms of the age from which they can be introduced and in terms of type of plant-based drink. It is well known that plant-based drinks are nutritionally different from cow’s milk, and they cannot be conceived as a simple replacement for cow’s milk.

It is precisely highlighting these differences that we suggest different applications in different situations. Personalized management of the choice is essential, based on an evaluation of the use in the specific case according to the chosen product, the health status of the child, and the context of the rest of the child’s diet.

Finally, clearer and more detailed labelling can help consumers make a better choice.

## Figures and Tables

**Table 1 foods-12-01544-t001:** Energy and macronutrient composition of infant and follow-on formulas [39,40] *.

	Energy (Kcal/100 mL)	Protein (g/100 Kcal)	Lipids (g/100 Kcal)	Carbohydrates (g/100 Kcal)
Animal-milk formula	60–70	1.8–2.5	4.4–6	9–14
ISP formula	60–70	2.25–2.8	4.4–6	9–14
HP formula	60–70	max. 2.8	4.4–6	9–14
Full-fat cow’s milk	60	3.3	3.3	4.6

* ISP: Isolated-soy-protein formula; HP formula: hydrolyzed-protein formula; see text for details.

**Table 2 foods-12-01544-t002:** Energy and macronutrient composition of plant-based drinks and cow’s milk [11,40,42].

	Energy (Kcal/100 mL)		Protein (g/100 mL)		Lipids (g/100 mL)		Carbohydrates (g/100 mL)	
Soy	44 *	46.7 §	3.3 *	3.1 §	2.0 *	1.8 §	3.0 *	4.3 §
Rice	57 *	56.8 §	0.2 *	0.3 §	1.0 *	0.9 §	12.0 *	11.5 §
Almond	38 *	40.2 §	0.8 *	0.8 §	2.3 *	2.0 §	3.0 *	4.4 §
Oat	47 *	45.3 §	0.6 *	0.9 §	1.2 *	1.1 §	7.9 *	7.5 §
Full-fat cow’s milk	60		3.3		3.3		4.6	

* Median value; § mean value; see text for details.

**Table 3 foods-12-01544-t003:** Suggestions for the use of formulas in cow’s-milk protein allergy (CMPA) [9,16,22,24,25,50,51].

Situation	<6 Months	≥6 Months
Non-severe CMPA	EHF Rice	EHF Rice Soy *
Severe CMPA, such as: Anaphylaxis FPIES Multiple allergies Severe enteropathy	AAF	AAF
CMPA, in the presence of symptoms not resolved by EHF	AAF Rice	AAF Rice Soy *
CMPA, in the presence of problems with EHFs	AAF Rice	AAF Rice Soy *
CMPA, in the presence of problems with AAFs	Rice	Rice Soy *

CMPA: cow’s-milk protein allergy; EHF: extensively hydrolyzed formula; AAF: amino-acid formula; FPIES: food-protein-induced enterocolitis syndrome. * In the absence of gastrointestinal symptoms.

**Table 4 foods-12-01544-t004:** Suggestions for the use of plant-based milk alternatives in specific situations starting from 1 year of age.

Situation	Choice
Obesity Overweight Dyslipidemia	- Soy - Almond - Oat - Spelt - Limit rice - Limit cow’s milk, prefer semi-skimmed milk - Formulas: 12–36-months
Types 1 and 2 diabetes mellitus	- Soy - Oat - Limit rice
Poor weight and height growth	- Cow’s milks - Soy - Rice *
CMPA	- Milk formulated for CMPA - Plant-based drinks *
Lactose intolerance	- Lactose-free cow’s milk - Plant-based drinks *
Family preference	- Plant-based drinks *

CMPA: cow’s-milk-protein allergy. * A limitation on the use of rice-based drinks in the first years of life is recommended. Note: The choice for a specific product should be made in agreement with a qualified professional (see text).

**Table 5 foods-12-01544-t005:** Suggestion for the use of plant-based drinks.

Age	Cow	Soy	Rice	Oat	Almond	Other
12–24 months	150 mL Whole Suitable as such	200 mL Suitable as such *	200 mL Provide protein and fat in the meal	200 mL Provide protein in the meal	200 mL Provide protein and fat in the meal	Evaluate on a case-by-case basis
>24 months	200 mL Partially or totally skimmed Suitable as such	200 mL Suitable as such	200 mL Provide protein in the meal	200 mL Provide protein in the meal	200 mL Provide protein in the meal	Evaluate on a case-by-case basis

* Possible use of a source of fats in addition to those derived from soy.

## Data Availability

Data is contained within the article.

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
