# Peer review of "Plant-Based Milk Alternatives in Child Nutrition"

_foods, 2023, doi:10.3390/foods12071544_

Round 1

Reviewer 1 Report

Overall, the topic is interesting and meaningful. The figures are appropriate and help the understanding. The manuscript is well organized and has sufficient literature research.  

Line 173, the title should be revised, such as choosing beverages for diet supplementation/nutrition reasons. The manuscript did not focus on medical reasons, so the presence of “non-medical” reasons in the title seems abrupt.

Line 231-243, This section discussed why choosing plant-based milk beverages, however, the authors used a long paragraph to explain why plant-based milk substitute was not equivalent to cow’s milk. This clarification is meaningful, but can be shorten or stated in another independent section.

Line 261-280 The authors compared the guidance/suggestions of plant milk consumption between different countries. However, the title was “why choosing plant milk”. The comparison of guidance can be put in a separate section. In addition, this section is a very long paragraph and should be divided into several paragraphs so it is easy to read.  

Line 329, Half of this section talked about the recommendation for cow’s milk consumption; it can be shortened or the authors need to explain the relationship between cow’s milk consumption and plant milk consumption. If 150 ml of cow’s milk is recommended for 1-2 years of age, how much plant milk will recommend for 1-2 years of vegan children? Is there any way for this conversion or calculation? I think it is more beneficial if the authors discuss how to determine the recommendation quantity of plant milk for children at different ages.

Line 367, It should be noted that many of the commercial plant milk beverages have been added with sucrose (or other types of sweeteners), so the calories and the total carbohydrates are higher than the whole milk. Even though the authors have discussed this limitation in the latter part of their manuscript, they should clarify the plant milk discussed in this section is sugar added or non-sugar beverage.

Line 460, The general advice should correlate with the weight of the children. For example, the recommended quantity of certain plant milk should be expressed as XX mL/g weight. In addition, the recommendation guidance commonly varied with different areas, diet, average weight and so on.

I am wondering the opinions from the parents (consumers). Except the vegan family and children with allergies, do parents show positive attitude to plant milk? Do they think the plant milk have high nutritional value and would like to add plant milk to the diet of their children?

Reviewer 2 Report

The title is misleading. It is written as White Milk-Like Plant-Based Beverages and Children: from the Literature to a Personalized Approach, but in the PDF file, it is written as Milk-like plant-based beverages in children nutrition. However, both titles are incorrect even on the name of the beverages: milk-like plant-based or plant-based milk beverage. It depends on what types of plant foods are covered as not all plants could make into milk-like beverages. It is also very subjective to define what is milk-like beverage, often just based on their physical and sensory characteristics. Thus, to avoid industry/public misinterpretation, these technical terms are regulated in most countries.  

The title does not seem to reflect the contents of the manuscript. 

There are so many types of plants (grains/cereals, beans/legume, nuts, seeds, etc.) that could be made into beverages by further processing. However, their nutritional compositions are diverse and vary greatly. Thus, what plant-based milk are referred here must be made known, otherwise their contribution to the dietary intake and public health could be very different.  

Most of the contents are not related to the scope of the title. The discussions are not about the contribution/implication of plant-based milk beverages in children nutrition. In addition, some of the sections (4 and 5) are deviated from the title.  

The roles of these plant-based beverages on children with different age groups are the similar, except with different levels dietary requirements. Thus, it is very obvious that section 2 and 3 are redundant. In addition, it should not be divided based on children ages but probably their contributions and implications on children nutrition and health.

The nutritional quality of plant-based protein is incomparable to those of animal protein. Thus, it is unlikely these plant-based milk beverages (except soymilk) could be used as a substitute for animal milks according to their nutritional composition. They could be just supplementary. 

A review is not limited to provide a summary of the past studies. It should critically examine the current state-of-the-art and express informed views, identify the gap of knowledge (or challenges) and provide direction for potential future research.

Reviewer 3 Report

The manuscript is a modified version of the titled manuscript „White Milk-Like Plant-Based Beverages and Children: from the Literature to a Personalized Approach”. In my opinion, the current version is better and corresponds more to the structure of the literature review. In my opinion, however, such a literature review should be more critical and analytical in terms of the data presented, taking into account the diversity of information collected. Hence my suggestion that the authors revise the text of the manuscript.

In my opinion, the graphic summary should be changed. The label “food allergy” is shown on it, which indicates that plant-based milk-like beverages are generally free of food allergens. Meanwhile, the manuscript only mentions allergies to milk proteins. It is also known that plant-based beverages are not allergen-free, as they contain many food allergens of plant origin. I suggest changing the graphic summary to “milk allergy”.

Lines 71-72 – that's what you write: „As plants are completely (and naturally) lactose-free, these products are proposed as alternatives to cow's milk [15,18].”. Meanwhile, there is no recommendation mentioned in the cited reference [18]: “This study showed that PBMS’s differ remarkably in nutritional and physicochemical properties. Depending on the raw material, some had very low protein contents and high glycaemic values. If these products are portrayed as cow’s milk substitutes, the nutritional inferiority can cause severe illnesses. To the authors’ knowledge, this paper presents the first assessment of many PBMS’s, taking several nutritional values into account. Especially the determination of GI values gave new insights to evaluate the nutritional importance.”. I suggest carefully analyzing the literature and checking the information contained in the manuscript so that it does not represent an over-interpretation of reference data.

In my opinion, chapter “6. Limitation” should go into more detail on the nutritional differences between milk and plant-based beverages in more detail. It is not just about the protein, fat or calorie content, but also about the nutritional value of protein, the content of deficient amino acids, minerals and vitamins. How are these aspects related to the daily nutritional needs of the child's organism?

In my opinion, chapter “7. Conclusions” needs to better correspond to the stated purpose of the manuscript.

Reviewer 4 Report

The manuscript submitted for review raises interesting issues related to the use of milk-like plant-based beverages in child nutrition.

However, I have some remarks which are as follows:

Some information concerning mechanisms of allergy to cow's milk proteins, galactosemis and lactose intolerance in children should be provided.

Point 2, and especially 3 - part discussing non-medical reasons is too extensive compared to other ones.

Generally, the authors describe rather various opinions concerning choosing of milk-like plant-based beverages than scientific opinions. Therefore, despite the interesting premise, I feel somewhat unsatisfied about the scientific aspect of the review.

Other comments:

replace "and colleagues" with "et al." in the whole text

line 190 - some reference concerning vit. D3 production from lichens should be added

line 374 - "....mono- and polyunsaturated" - it looks as if something was missing here

lines 511-512 - What kind of morning physical activity in very young children?

Round 2

Reviewer 2 Report

The title does not seem to reflect the content of the manuscript, especially on the advice and food characteristics. 

Please define clearly what are plant-based milk substitutes in the abstract/introduction. Not all plant-based products could be considered as milk substitutes. They should have some similar nutritional composition like a cow milk then only could be claimed as an animal milk substitute.  

Avoid using 'etc.' in the text as this is meaningless in scientific discussions.

Provide a table comparing the nutritional characteristics of the respective plant-based milk substitutes (soymilk, rice, oats) with the cow's milk.

Plant-based milk substitutes and plant-based beverages are two different products defined by law and regulations in many countries. They are not the same products. Thus, their names (plant-based substitute and plant-based beverages) should not be used interchangeably in the manuscript. 

The roles of the plant-based milk substitutes on children groups (different by ages) are similar, probably their level of dietary requirements could be differed. This is why the subtitles for section 3 and 4 are so similar. Thus, there is no need to separate it into children with <12 months and children with >12 months, as many of the subtitles are repeating. You should write the impacts of these plant-based milk substitutes on children nutrition and health, the contributions and effects of plant-based milk substitutes on general children nutrition and health. 

Therefore, the authors are expected to re-structure the subtitles and re-arrange contents of section 3 and 4 into one section.   

The contents of any table or figure must be discussed in the text. Please do not just list down the table/figure but without any in-depth discussions.

The title for Table 1 does not seem to reflect the content. Formula is a manufactured food (generally cow's milk) designed and marketed for feeding babies and infants under 12 months of age. This term (formula: infant formula or baby formula) is regulated by legislation in most countries. No product could be called formula unless the product could meet the criteria set by the regulations.  

Table 2 is misleading. The choices and situations are superficial. It does not reflect the previous works done. You cannot request someone to consume soy or almond or oat as a choice to overcome obesity. 

Section 5 - General advice on what? 

Section 6 - This section tends to repeat the similar situations that have been mentioned in section 3-4. 

The perspective of future research does not seem to be an outcome from this review (based on the issues or gap of knowledge reported in this review).

Based on the above comments and the incorrectly used of technical terms, especially those regulated by food laws or food regulations (as complicated by the differences in between countries). 

Reviewer 3 Report

I see that the Authors have accurately answered the questions posed by the reviewers. I am satisfied with these answers and explanations. I also have no objection to the revised version of the manuscript.

Author Response

The authors thank Rev#3 for the comments 

Reviewer 4 Report

I am satisfied with the authors' corrections and explanations.

Author Response

The authors thank Rev#4 for the comments